# Psycho-social factors associated with type two diabetes remission through lifestyle intervention: A scoping review

Kathy J. Hart[o], Andrew Kubilius[o], Martin Clark[iD][o]*

Department of Psychology and Computer Science, University of Central Lancashire, Preston, Lancashire, United Kingdom

[o] These authors contributed equally to this work.
* MClark11@uclan.ac.uk

## Abstract

### Objectives

There is growing evidence that type 2 diabetes (T2D) can be put into remission through lifestyle intervention. Current focus on remission in terms of physiological considerations and biomedical scales, means there is limited understanding of the role psycho-social factors play in moderating the efficacy of lifestyle interventions for T2D remission. In the current review we aimed to synthesise the emerging literature on psycho-social factors associated with T2D remission, specifically from lifestyle interventions.

### Methods

Five databases (EMBASE, MEDLINE, CINAHL ultimate, PsychINFO and PsycArticles) were searched to identify studies from 2009 onwards that reported remission outcomes from lifestyle interventions in participants $\geq$ 18years old, with a clinical diagnosis of T2D. Studies included were of an interventional or observational design and restricted to English language. Screening and data extraction was performed independently by two reviewers using prespecified criteria.

### Results

In total 6106 studies were screened, 36 studies meeting the inclusion criteria were included. Studies were globally diverse, with 30 (83%) being published $\geq$ 2017. Psycho-social scales were under-utilised with 22 (67%) of studies failing to include any psycho-social measures. Single arm, prospective studies were most frequently utilised, however study quality was perceived to be heterogeneous. Further disparity in the quality, content and delineation of the psycho-social interventions was also identified. Education and self-monitoring interventions were the most frequently incorporated. Self-monitoring was also identified as an important facilitator to remission, in addition to social support.

**Data Availability Statement:** All relevant data are within the paper and its Supporting information files.

**Funding:** The author(s) received no specific funding for this work.

**Competing interests:** The authors have declared that no competing interests exist.

## Conclusions

Our findings indicate that psycho-social factors in T2D remission are under-explored. We have identified a number of methodological issues (comparability, conflicting remission criteria and poorly defined psycho-social interventions) suggesting knowledge gaps which could inform the methodology of future research design. There is significant opportunity for future research to incorporate the social model of disease, conceptualise remission more holistically, and build a more comprehensive evidence base to guide clinical practice.

## Introduction

In 2021 an estimated 536 million of the global population were affected by type two diabetes (T2D) [1] Saeedi and colleagues [2] estimate this will rise to 700 million by 2045. Physiologically individuals with T2D are at risk of multiple serious morbidities, whilst psychologically, depression, anxiety, diabetes distress and diabetes stigma are all prevalent and associated with poorer health outcomes [3,4].

There has been a paradigmatic shift in how T2D is conceptualised and treated [5]. Historically, it was considered a chronic condition requiring increased medication as it progressed [6–8]. Recently, evidence from bariatric studies, and from dietary and lifestyle interventions, has now elucidated that with sufficient weight loss, remission from diabetes could be an achievable outcome [9–12]. Diabetes remission has been established as a primary research priority by the James Lind Alliance [13,14], however this remains an emergent field with significant heterogeneity as to how remission and the associated clinical parameters should be delineated [3,15–18].

In addition to a lack of consensus regarding a consistent definition of diabetes remission, there is also a lack of scrutiny of the psycho-social factors which could moderate or mediate remission outcomes. 'Psychosocial' is defined by Martikainen et al. [19] as "the influence of social factors on an individual's mind or behaviour, and to the interrelation of behavioural and social factors". These factors play a key role in mediating the effects of social determinants on an individual's health, impacting physical health directly as protective or risk factors, and also through their influence on health behaviours such as physical activity levels and diet [20]. In diabetes management research e.g. Tharek et al [21] has already established the critical role of multiple psycho-social factors in self-care and glycaemic control. Self-efficacy [21–23], social support [24–28], digital health literacy [13,29,30] and motivation [14,31,32] have all been found to positively impact health outcomes. There are multiple psycho-social scales, both generic and diabetes specific, e.g. Polonsky et al [33,34]; Bijl et al [35], which have been developed to measure the impact of these factors and aid the delivery of person-centred care but it is unclear how widely or frequently these have been applied in remission studies.

Within current consensus statements [18] remission is narrowly focused on "physiological considerations and monitoring of biomedical indices", to the detriment of our understanding of psycho-social factors [36] (p 406). Multiple studies, e.g. Nicolucci et al [37]; Whicher et al [38] and Wylie et al [39] also identify that the psychological and emotional needs of individuals with T2D are consistently an unmet priority. This limits our understanding of the support individuals pursuing remission may require. It also means that remission, in the absence of appropriate tailored support, may negatively impact an individual's physical or mental health [3,36].

The primary research objective is to identify the psycho-social factors associated with T2D remission through lifestyle intervention. The secondary objectives are to identify the psycho-social scales used within remission studies; assess the method and design of these studies; identify the psycho-social interventions incorporated; determine how the reported psycho-social factors may influence remission outcomes and finally to identify gaps within the evidence base. It is hoped this review will inform future research design and aid practice of Health Care Professionals (HCP) working with individuals with T2D in identifying the potential barriers and facilitators to remission.

## Methods

This scoping review was conducted following guidance from The JBI Manual for Evidence Synthesis [40], Peters et al. [41] and Arksey and O'Malley [42]. Reporting is based on the Preferred Reporting Items for Systematic reviews and Meta-Analyses extension for Scoping Reviews (PRISMA-ScR) [43], with the completed PRISMA-ScR being provided in supporting information (see S1 Table). The objectives, methods and inclusion criteria for this review were all determined *a priori* and registered on the Open Science Framework (OSF) prior to literature screening: https://osf.io/pu3k4/?view_only=7661ed50e1fe4f31afe53c71239ce9c7.

### Eligibility criteria

In line with the JBI manual (2020), inclusion criteria were formulated according to the Population, Concept and Context (PCC) mnemonic, see Table 1. This establishes the scope of the review in identifying the desired study population (adults with T2D), the phenomenon of interest (remission) and any specific restrictions. Of fundamental importance to comparability was the need for authors to include remission outcomes according to whichever criteria they specified. This is a complexity inherent in this field of study where neither a descriptor nor consensus to remission criteria has been agreed. Studies were restricted to 2009 onwards when the first preliminary discussion on remission descriptors and criteria, was published [15]. No geographical restrictions were imposed. Participants under the age of 18 were excluded as is common within existing diabetes remission reviews (see Captieux et al [44] and Goldenberg et al [45]). The focus on this review is on the psycho-social domain including behavioural change through lifestyle or health behaviour intervention. Studies included were interventions that sought to change health related behaviours, diet or exercise, in the absence of new pharmaceutical agents or surgery.

**Table 1. Population, Concept and Context (PCC) inclusion criteria [46].**

| PCC | |
|---|---|
| **Population** | **Population: T2DM**<br>**Include**: pre-existing clinical diagnosis of T2D, adult<br>**Exclude**: patients who 'self-identify' with 2TD, diagnosis through screening within study, < 18 years old, any other type of diabetes (e.g., type one, gestational, MODY), combined study population of pre-diabetes and T2D where results cannot be extracted separately |
| **Concept** | **Concept: Remission**<br>**Include**: diabetes remission through health behaviour intervention<br>**Exclude**: remission from diabetic complications, studies where remission outcomes are not reported, remission through surgical or pharmaceutical means<br>**Outcome**: psycho-social scales, psycho-social interventions, psycho-social factors reported by participants |
| **Criteria** | **Context**:<br>**Include**: interventional or observational studies, ≥ 2009, English language, any geographical area<br>**Exclude**: secondary literature (reviews, commentary pieces etc.) |

### Search strategy and information sources

Preliminary searches of CINAHL and MEDLINE were conducted with a Senior Librarian at the University of Central Lancashire (UCLan) and a search strategy developed *a priori* in June 2022. Key search terms were extracted from consensus remission statements [16–18] (see S2 Table). The electronic databases EMBASE, MEDLINE, CINAHL ultimate (see S3 Table), PsychINFO and PsycArticles were searched in September- October 2022. Key terms were added verbatim to each database with minor adjustments made to index terms and database specific terminology where necessary. Additional records were identified through web searches and hand searching reference lists. Grey literature was searched using Google Scholar.

### Data management and software

MS Excel was used to calculate statistical data and compile frequency charts. References were collated, uploaded and de-duplicated using the reference managing software EndNote X9 [47].

### Study selection

Two reviewers independently screened all titles, abstracts and full texts excluding studies that did not meet the inclusion criteria. Any discrepancies in inclusion were resolved firstly through discussion. Where we were unable to achieve a consensus, the text was referred to a third reviewer for independent evaluation.

### Data extraction

Data was independently extracted and summarised within two data extraction forms, developed *a priori*. The forms were piloted using a sample of three studies [48–50]. The first form (see S4 Table) focused on data pertaining to study design, psycho-social scales, and the impact of psycho-social factors on remission. The second form, on the psycho-social interventions (See S5 Table).

### Data analysis

Studies using the following designs: random crossover trials, open-label cluster randomised trials and open-label, parallel-group and randomised controlled trials (RCTs), were grouped together as RCTs for analysis. For the psycho-social interventions, studies that linked to the same RCTs, and therefore the same interventions, were also grouped together. Results for both interventional and observational studies were synthesised in the same manner. In line with the scoping review methodology, risk of bias has not been formally evaluated. However, indication has been made to potential quality issues, data inconsistency and where further research should be directed to address deficits.

## Results

Initial database searches identified 6100 records, with a further six identified through hand searching reference lists. Searches for grey literature did not yield any additional records. After de-duplication 5038 records remained, limited to 141 after title and abstract screening. After full text screening, 36 records were identified which met the inclusion criteria, see Fig 1. Of the studies included 30 (83%) were published ≥ 2017, reflecting the emergent nature of the field. The sample was globally diverse, with studies being conducted in UK (10), USA (9), India (3), Italy (2), Thailand (2), Barbados (1), Canada (1), Brazil (1), the Netherlands (1), Switzerland (1), Israel (1), Denmark (1), Quatar (1), South Africa (1) and China (1).

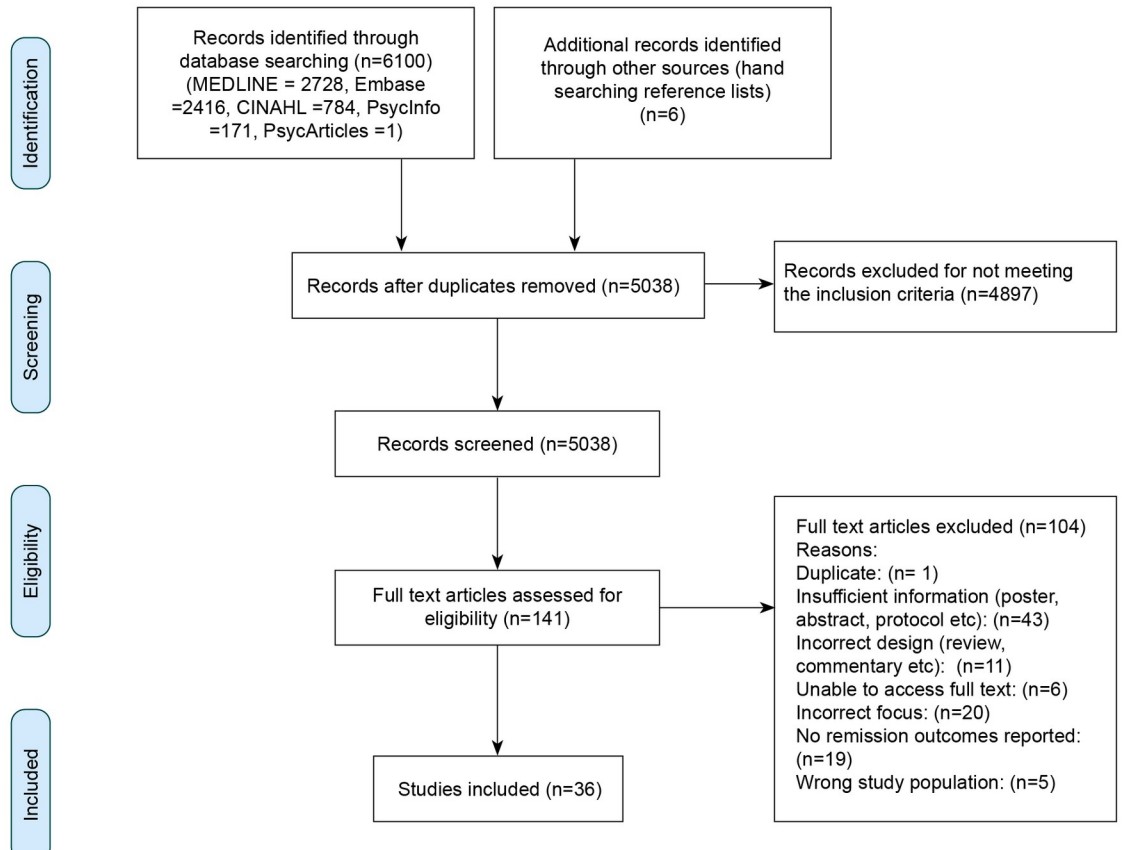

**Fig 1. PRISMA flow diagram for the electronic search and screening process [51].**

## Psycho-social scales

Analysis of the 33 studies which were quantitative or mixed method (qualitative studies due to their nature were unlikely to include psycho-social scales and were therefore excluded) revealed that 22 (67%) failed to include any psycho-social measures (see Fig 2). Of the studies which did include psycho-social measures, the majority were limited to one quality of life scale, either the 36 item Short Form Health Survey (SF-36) or the EuroQol (EQ-5D) battery. Only three studies utilised diabetes specific scales [52–54].

## Study designs

Of the 36 studies included, 13 (36%) were retrospective in design (Fig 3). Where the study involved retrospective analysis of an intervention, with the exception of the qualitative studies, the design of the original study was used to inform findings. Particularly significant Randomised Controlled Trials (RCTs) such as the seminal DiRECT study have generated multiple primary [11,12] and secondary [8] analyses. RCT's (n = 10) comprised 36% of the sample. Single arm prospective studies (n = 13), usually pilot or feasibility with n≤45, were the most utilised study design (50%).

Whilst participants in all reviewed papers had an existing clinical diagnosis of T2D, study populations were otherwise heterogeneous [55]. Restrictions to inclusion criteria in the papers reviewed were based on multiple factors: duration of diagnosis, being non-insulin taking and/

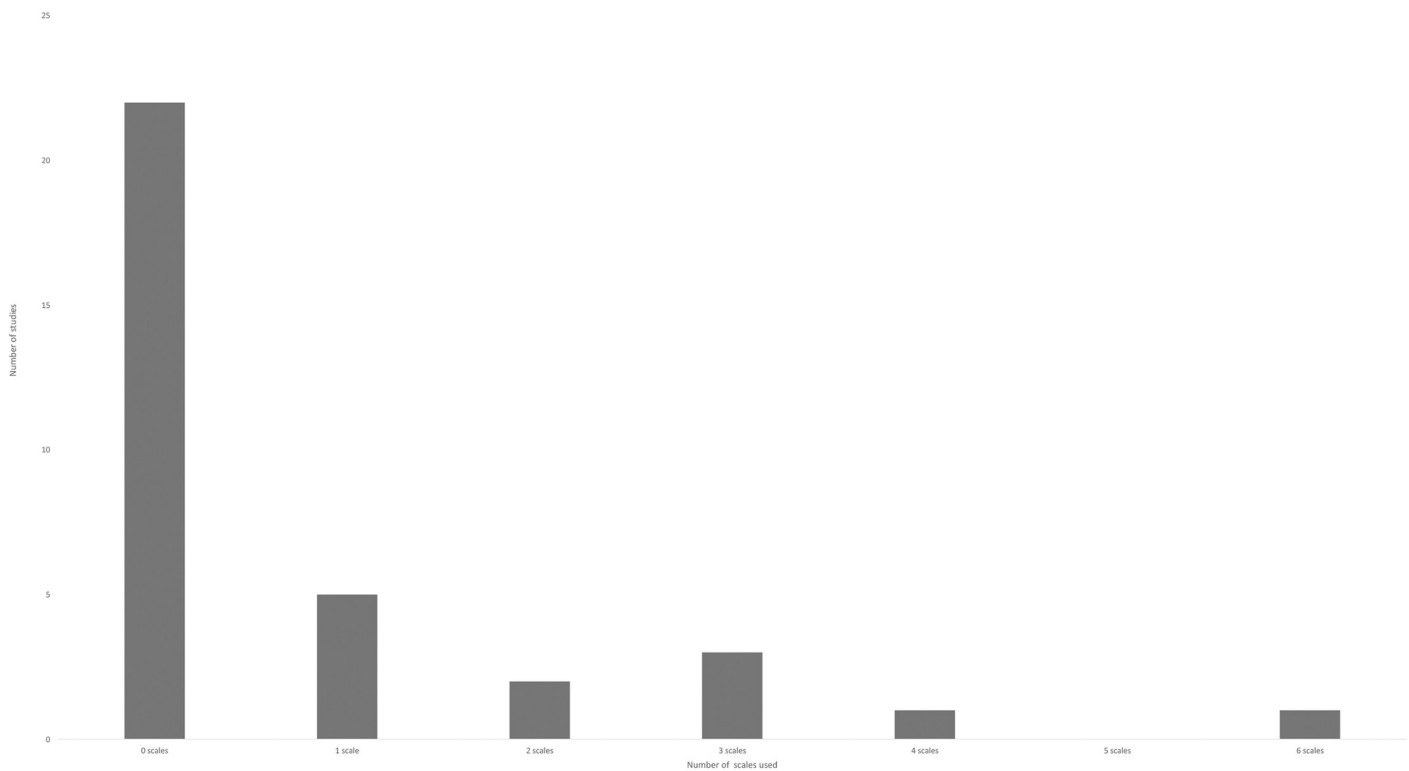

**Fig 2. The number of psycho-social scales utilised in all studies reviewed.**

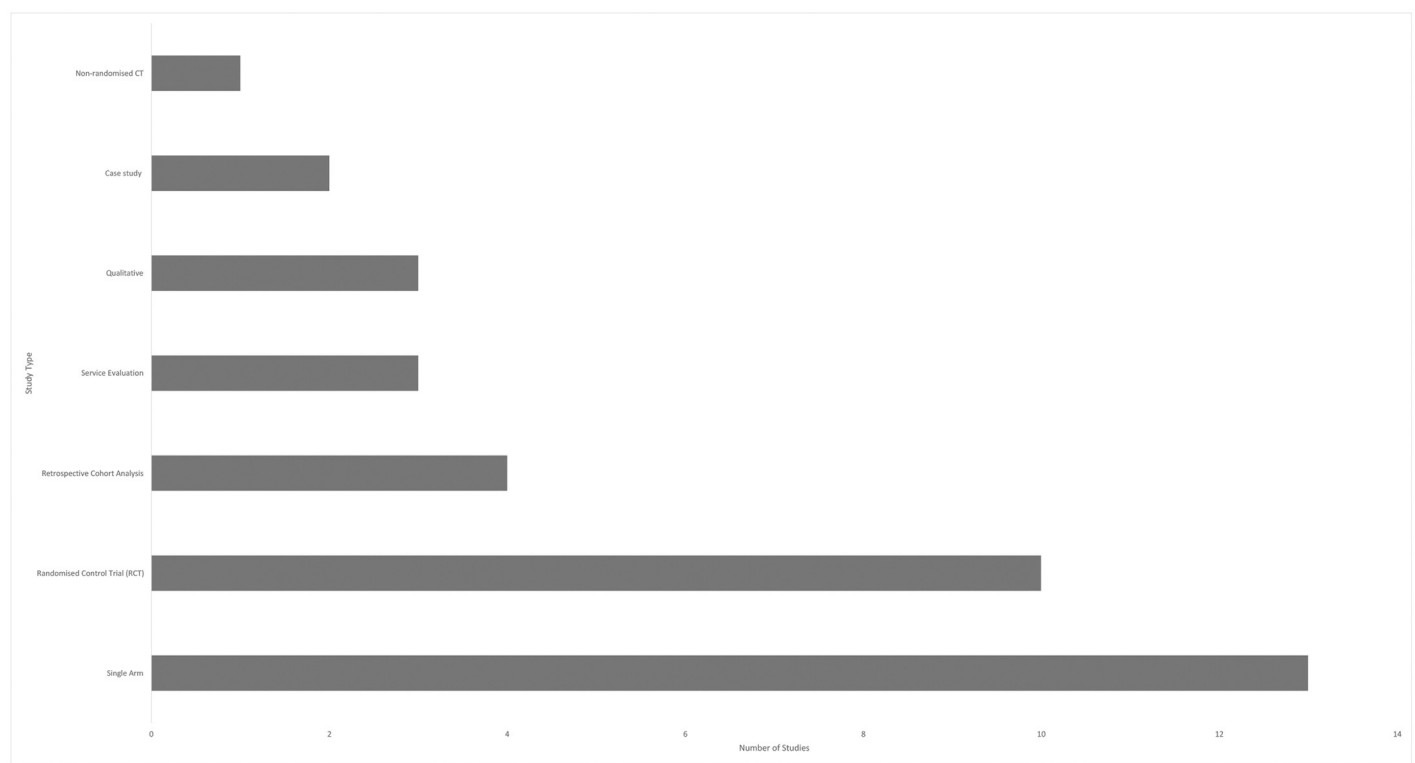

**Fig 3. The number of studies reviewed that confer to one of the seven designs.**

or naïve to glucose lowering medication, age, existing diabetes complications, existing mental health diagnosis, BMI and HbA1c level.

## Psycho-social interventions

Interventional studies were grouped and analysed (n = 27), Evans et al. [56] was used to guide identification of intervention specifics. Insufficient detail within the papers necessitated scrutiny of clinical trial registrations, protocols and other cited papers, however in some cases these were inaccessible as indicated on Fig 4. A lack of precision in terminology or the use of generic expressions such as 'care', 'support' and 'counselling' in the absence of context or further detail, resulted in a lack of clarity indication.

**Key: + signifies evidence of the intervention,—no evidence of the intervention, ? lack of clarity around the intervention and \* unable to access protocol.** There was a clear disparity in the content of the psycho-social interventions, with some offering no evidence of any psycho-social intervention (n = 4), and others being multifaceted incorporating 5 or more features (n = 9) (Fig 4). The most commonly used intervention was education (n = 18); however, it was also the most under-described. Self-monitoring (n = 16), the next most frequently adopted intervention, was more explicit in both the measures used (CGM, food diary, weight checks, accelerometers) and how these impacted other dimensions such as goal setting. There was evidence of theoretical underpinning or supportive studies being cited in (n = 11).

## Influence of psycho-social factors on remission outcomes

Whilst there is only limited quantitative data from psycho-social scales presented in the reviewed papers, emergent positive relationships were apparent. With participation in the interventions(in which some achieved remission and the majority demonstrably improved their condition), being associated with increased quality of life [8,11,12,57–59], increased self-efficacy [60], significantly increased self-esteem [52], increased work and social adjustment [52], reduced binge eating [8,52], reduced problem areas in diabetes management [54] and finally increased diabetes knowledge and empowerment [53]. There were inconsistencies in relation to anxiety and depression [52,57,60,61].

From the qualitative and mixed method studies, the most prominent factor was the role of social support highlighting both a facilitatory [62] and inhibitory influence on remission outcomes [63]. Where the support was positive and continuous, participants commented that this was beneficial to motivation, accountability and adherence. However, where support was lacking or inconsistent, participants noted the significant challenges of navigating negative feedback and social saboteurs [63,64]. A further key observation related to perceptions of self-image and identity, and how this shifted in relation to weight loss and the appraisal of others, [63,65]. A final shared observation related specifically to the shifting cognition associated with total diet replacement (TDR) and subsequent food reintroduction and included themes relating to decision making, self-regulation, adaption and autonomy [63,65,66].

## Discussion

In this scoping review we analysed 36 studies based on lifestyle or health behaviour intervention which reported T2D remission outcomes. To our knowledge this is the first review to specifically address the psycho-social factors associated with remission and broaden the focus from the dominant bio-medical perspective.

Our findings confirm that psycho-social factors in T2D remission are often overlooked [36] or receive minimal acknowledgement. Our review indicated that only 11 out of 33 interventional studies included psycho-social scales and 23 out of 27 studies employed psycho-social

| Intervention | Self-monitoring | Goal setting | Cognitive restructuring | Education | Individual sessions | Group sessions | Peer support online | Involvement family/friends | Cited studies/ theoretical underpinning |
|---|---|---|---|---|---|---|---|---|---|
| **Study** | | | | | | | | | |
| Ades et al (2015) | + | + | + | + | ? | + | - | - | + |
| Athinarayan et al (2019) | + | ? | ? | + | ? | ? | + | - | + |
| Bhatt et al (2017) | + | - | - | ? | + | - | - | - | - |
| Bynoe et al * (2019) | - | - | - | - | - | - | - | - | - |
| Correa et al (2022) | - | - | - | + | - | - | - | - | - |
| Cox et al (2019) | - | ? | ? | + | + | - | - | - | - |
| Dave et al (2019) | + | ? | ? | + | + | - | - | + | - |
| De Hoogh et al* (2021) | + | ? | ? | + | + | - | - | - | - |
| Esposito et al (2014) | - | - | - | + | - | - | - | - | - |
| Gregg et al (2012) | + | + | + | + | + | + | - | - | + |
| Lagger et al (2018) | ? | + | + | + | + | + | - | - | + |
| Lean et al (2018), Lean et al (2019), Rechakova et al (2021) Thom et al (2021) (DiRECT study) | + | - | - | + | + | - | - | - | - |
| Marples et al (2022) | - | + | ? | + | + | + | - | - | + |
| Mottalib et al (2015) | + | + | + | + | ? | + | - | - | + |
| Oser et al (2022) | + | ? | + | + | - | - | - | - | + |
| Rechakova et al (2017) Rechakova et al (2019) Steven et al (2016) (Counterbalance study) | + | + | ? | + | + | - | - | - | + |
| Rein et al (2022) | + | - | - | - | ? | - | - | - | - |
| Reid-Larsen et al (2019) | + | - | - | + | ? | + | + | - | + |
| Romana et al (2019) | - | - | - | - | - | - | - | - | - |
| Sarathi et al (2017) | - | - | - | - | - | - | - | - | - |
| Taheri et al (2020) | + | + | + | + | + | - | - | ? | + |
| Umphonsathien et al * (2019) | + | - | - | - | - | - | - | - | - |
| Umphonsathien et al* (2022) | + | - | - | - | ? | - | - | - | - |
| Unwin et al (2020) | - | + | + | + | + | + | - | + | + |
| Webster et al (2019) | - | - | - | - | - | - | - | - | - |
| Yancy et al (2019) | + | ? | ? | + | - | + | - | + | - |
| Zou et al (2022) | - | - | - | - | + | - | - | - | - |

**Fig 4. Evidence of psycho-social interventional features in all the studies reviewed.**

interventions, although in some instances these only just met the inclusion criteria. There was significant study heterogeneity and a number of comparability issues: variable trial quality, conflicting remission criteria, differences in remission rates and variability in study populations. Furthermore, psycho-social interventions tended to be poorly delineated which made analysis more challenging.

Psycho-social scales were underused and indeed absent in the majority of studies included. The average number of biomedical scales utilised per study was 12.15, in comparison psycho-social scales averaged at only 0.76. Where psycho-scales were utilised the most frequent were SF36 scale or the EQ-5D battery which measure quality of life. However, these are non-diabetes specific, and the latter has been subject to criticism for its generic nature and insufficient focus on mental and social aspects of health [67]. Effectively they give little indication to participant's cognitive, emotional or behavioural states, beyond a general concept of 'wellness'. As psycho-social factors are critical both in diabetes management [13,22,25,68] and weight loss [69] (the greatest predictor of remission) their underuse within these studies, means there is little quantitative evidence base to identify barriers or facilitators of remission. The inclusion of a broader number of scales at baseline, for example in motivation, self-efficacy or self-esteem, could potentially identify those who may find behavioural change more challenging and suggest potential for preliminary intervention such as motivational interviewing [70] or self-efficacy training [71] both of which have proven efficacy in promoting behaviour regulation. Clearly though this is still an emerging field, and as time progresses there is evidence of greater scrutiny of psycho-social factors. In the studies we reviewed between 2012 and 2017, psycho-social scales were completely absent. From 2018 onwards there has been a slow progression to increased inclusion.

The presence of psycho-social scales within the retrospective studies (n = 13) was dependent and, in many cases, limited by the design of the initial research. This was particularly apparent in the retrospective cohort studies reliant on routine clinical data, as psycho-social measures are rarely utilised in primary care despite these factors being an essential component of person-centred care [72]. There are also known quality and methodological issues inherent with the use of retrospective cohort studies [73] and case studies [74,75]. The most frequent study design, single arm (n = 13/36), measured the initial efficacy of a lifestyle intervention, however the lack of a control arm means that spontaneous recovery or placebo effects cannot be ruled out in relation to remission or psycho-social factors [76]. Furthermore, as the majority of participants either self-selected or in some instances were subject to a compliance check prior to participation [58] study populations may have had higher motivation levels than the general diabetes population.

Randomised Control Trials (RCTs) are widely considered to be the gold standard in research due to the fact they provide more rigour in examining the relationship between intervention and outcome, in addition to minimising bias [77]. Perhaps the most conclusive of these in terms of the relationship between psycho-social factors and remission is the RCT by Taheri et al. [57] due to the higher number of those achieving remission (61% in the intervention arm compared to 12% in the control group). However, within the intervention arm of the included RCTs remitters and non-remitters were rarely separated. As a result, there could be a number of psychological processes underpinning remission that are not being explored. Existing studies have already indicated that duration of diabetes moderates ability to achieve remission which is usually attributed to beta cell dysfunction and pancreatic capacity [49,78–81], but the psychological processes implicit within this are under investigated. Steven et al. [81] note that remission may be more difficult to achieve for those who have failed at repeated attempts at weight loss. Arguably with longer duration of diabetes, coping mechanisms,

cognitive structures and eating behaviours will all have become more entrenched over time, resistance to change may therefore be psychological as much as it is physiological.

Analysis of the qualitative studies identified numerous psycho-social factors for example social support [82] and psychoeducation [83], which support pre-existing knowledge of facilitators to effective diabetes management. Our findings also suggest additional psycho-social factors uniquely associated with weight loss. Dietary restriction, particularly through TDR can result in dramatic weight loss over a short period of time, significantly impacting the appearance of the individual. It is suggested that this leads to challenges with perception of self and self-image [63,65]. It further invites more opportunity for negative societal comment, or social saboteurs [84]. Targeted psychological support may be required to support this process of physical and psychological change. A further facilitatory factor widely incorporated in the psycho-social interventions was self-monitoring (present in 16 of the 27 studies reviewed), specifically the role of biofeedback [62]. The use of CGM for example was evaluated by participants in Oser et al [53]as being the most useful component of the intervention. Whilst self-monitoring was identified as having an empowering effect [53] and there is literature to support the efficacy of both self-weighing [85] and self-monitoring blood glucose [86,87] there appears to be a degree of trepidation in health policy recommendations. The National Institute for Health and Care Excellence (NICE) [88] state that only those advised by a doctor or nurse should self-monitor their blood glucose. This may serve to disempower individuals who are pursuing remission in the community and is an important focus for further research.

Historically psycho-social interventions have been poorly delineated [89,90] this field proves no exception. Use of terms such as 'support', 'care' or 'counselling' without context or reference to theoretical underpinning complexifies analysis and threatens future replicability. In the majority of the studies lifestyle interventions were incorporated with some degree of psychosocial intervention, meaning the individual impact of either of these mechanisms cannot be extrapolated. This limits understanding of the optimum conditions required to support an individual to achieve and sustain remission. Consideration was given to the potential impact of Covid 19 on the nature and facilitation of the psycho-social interventions. We reviewed the end of each study's data collection period and whilst the majority were pre-Covid, Marples and colleagues [52]and Oser and colleagues [53] both discuss accessibility issues and digital adaptions due to the Covid restrictions. This is a pertinent point for future researchers to consider.

Furthermore, in a number of studies secondary focus/analysis was not on psychosocial factors, but instead rested on cost efficacy of the interventions and the prospects of savings for the respective healthcare services [78,91–93]. Focus on the financial implications of diabetes care may reinforce stigmatising rhetoric whereby the individual is blamed both for their own 'moral failure' to prevent what is perceived as a lifestyle disease and for the subsequent cost burden this incurs. There is a danger that remission could be seen as a panacea, a means of restoring normoglycaemia without costly pharmaceuticals or the need for sustained primary care and concerningly, Holman and colleagues [94] have recently indicated that once a patient is clinically coded as in remission routine diabetes care checks are reduced.

Strengths of this review include the globally diverse sample which minimises publication bias and the inclusion of a broad range of study types. These may previously have been overlooked by systematic reviews focused on diet with more rigid design exclusion criteria in place [95–97]. Clarity of process and reporting was achieved by adhering to both JBI [40] recommendations and the PRISMA-ScR [43] statement for scoping reviews. We adopted a systematic approach with all data extraction and screening being carried out independently by two reviewers. Limitations to this study include the restriction to English language, inability to access a number of protocols and the fact a more extensive search of grey literature could have

been implemented by accessing dissertation and theses repositories. Whilst under 18s were excluded from this review, given the increasing prevalence of T2D in young people and the paucity of existing research, this may represent an important focus for future research.

## Conclusion

This research aimed to investigate the psycho-social factors associated with type two diabetes remission through lifestyle intervention. Established as a research priority by the James Lind alliance and with significant interest from the diabetes community, this review has demonstrated that there is a clear need to address the psycho-social domain with renewed vigour. Currently, heterogeneity in study design and quality, variance in the study populations, a lack of consensus to remission criteria and poorly delineated terms are impeding replicability and the establishment of a sound knowledge base. For T2D remission to be implemented effectively within clinical practice, data from psycho-social scales and the evaluation of psychosocial interventions is critical to ensuring that advice and support offered to patients is both evidence based and person-centred. There are significant opportunities to tailor future research to reflect a more social model of disease which recognises the challenges of obesogenic environments, societal, family and cultural demands on individuals choosing to pursue remission. Rather than solely a bio-medical procedure, remission should be recognised, and treated, as a holistic process.

## Supporting information

**S1 Table. Preferred Reporting Items for Systematic reviews and Meta-Analyses extension for Scoping Reviews (PRISMA-ScR) checklist.**
(DOCX)

**S2 Table. Search terms used for the database searches of EMBASE, MEDLINE, CINAHL ultimate, PsychINFO and PsycArticles.**
(DOCX)

**S3 Table. Results from database search of CINAHL Ultimate (conducted 26/9/22).**
(DOCX)

**S4 Table. Data extraction form one- detailing study, country, participants, design, lifestyle intervention, number of psycho-social scales used and the effect of these on remission outcomes.**
(DOCX)

**S5 Table. Data extraction form two- detailing author and related studies, psycho-social interventions and cited studies which underpin the interventions.**
(DOCX)

## Author Contributions

**Conceptualization:** Kathy J. Hart, Martin Clark.

**Data curation:** Kathy J. Hart, Andrew Kubilius.

**Formal analysis:** Kathy J. Hart.

**Investigation:** Kathy J. Hart, Andrew Kubilius.

**Methodology:** Kathy J. Hart.

**Project administration:** Kathy J. Hart.

**Supervision:** Martin Clark.

**Visualization:** Kathy J. Hart.

**Writing – original draft:** Kathy J. Hart.

**Writing – review & editing:** Kathy J. Hart, Martin Clark.

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
