## [Decision Letter · Decision Letter 0]

29 Aug 2023

PONE-D-23-23765Psycho-social factors associated with type two diabetes remission through lifestyle intervention: a scoping reviewPLOS ONE

Dear Dr. Clark,

Thank you for submitting your manuscript to PLOS ONE. After careful consideration, we feel that it has merit but does not fully meet PLOS ONE’s publication criteria as it currently stands. Therefore, we invite you to submit a revised version of the manuscript that addresses the points raised during the review process.

We look forward to receiving your revised manuscript.

Kind regards,

Meisam Akhlaghdoust, M.D., M.P.H.

Academic Editor

PLOS ONE

Journal Requirements:

Reviewers' comments:

Reviewer's Responses to Questions

**Comments to the Author**

1. Is the manuscript technically sound, and do the data support the conclusions?

Reviewer #1: Yes

Reviewer #2: Yes

2. Has the statistical analysis been performed appropriately and rigorously? 

Reviewer #1: Yes

Reviewer #2: N/A

3. Have the authors made all data underlying the findings in their manuscript fully available?

Reviewer #1: Yes

Reviewer #2: Yes

4. Is the manuscript presented in an intelligible fashion and written in standard English?

Reviewer #1: Yes

Reviewer #2: Yes

5. Review Comments to the Author

Reviewer #1: Thank you, for undertaking this review which is timely and of topical interest. The restrictions criteria were interesting and I was unclear as to why under 18's were omitted from the review, this would have enhanced the reach and value of the review especially as the increase in young people developing Diabetes type two has accelerated during recent years and has been associated with obesity. Was there any obvious correlation with Lockdown and any psycho social interventions associated with life style factors evident in any of the reviews?

Some of your methods were lengthy and whilst robust may have the effect of losing readers interest.

Small issues include where you note in text on page 3 (72)"In diabetes management research say then straight away such as eg Burns, 2020 etc. Likewise, (76) Accordingly a number of studies ie , (86) multiple studies ie . In Eligibility criteria (113) please explain there briefly your three word inclusion ( its in appendix) but this would have explained earlier to me why under 18's were not included and not all readers will be familiar with every guide. I had to wait till (223) Restrictions to inclusion criteria to see no under 18's.

Reviewer #2: It is a practical paper in the field of diabetes and this is the first paper in this regard according to the knowledge of the authors. This paper is a well-written one and the authors adhered to the writing standards and valid JBI manual. However, there some comments as follows:

Overall, it is recommended to re-read the manuscript and shorten it. It is recommended to summarize the results as table or graphical format, which will be attractive for both researchers and clinicians.

Introduction:

- Please end the introduction section with primary and secondary objectives instead of questions. It is not common in scientific writing.

Methods:

- Line 126- 130, you mentioned “Table 1 adapted from: Breuing J, Pieper D, Neuhaus AL, Heß S, Lütkemeier L, et 127 al. (2020) Barriers and facilitating factors in the prevention of diabetes type 2 and 128 gestational diabetes in vulnerable groups: A scoping review. PLOS ONE 15(5): 129 e0232250. https://doi.org/10.1371/journal.pone.0232250. Modified and licensed 130 under the creative common attribution 4.0 international (CC BY 4.0).” It better to cite to this reference.

- How about searching for grey literature?

- In the abstract, you stated “ Studies included were of an interventional or observational design and restricted to English language”. However, in the method section, you did not mention anything about results synthesis of observational studies. I think all of the included studies were clinical trials.

Results:

- Line 195- 198 :” Fig 1. Adapted from: Page M J, McKenzie J E, Bossuyt P M, Boutron I, Hoffmann T 196 C, Mulrow C D et al. The PRISMA 2020 statement: an updated guideline for 197 reporting systematic reviews BMJ 2021; 372 :n71 doi:10.1136/bmj.n71. Modified and 198 licensed under the creative common attribution 4.0 international (CC BY 4.0)” Please cite to that reference.

- Line 201-202, you included 36 studies. But, you stated that “Analysis of the 33 studies which were quantitative or mixed method revealed that 22 202 (67%) failed to include any psycho-social measures”.

- It is recommended to summarize the results in a table. Most readers are not interested in reading the text.

Conclusion:

- Write this section based on the main research question.

6. PLOS authors have the option to publish the peer review history of their article (what does this mean?). If published, this will include your full peer review and any attached files.

Reviewer #1: **Yes: **Dr Toni Bewley

Reviewer #2: **Yes: **Mahin Nomali, MSCCN, Ph. D in Epidemiology

---

## [Author Response · Author response to Decision Letter 0]

27 Sep 2023

Response to reviewers

Dear colleagues, 

Thank you very much for your thoughtful and clear review comments. We have now worked through the submission and its figures. These have been fully redrafted in line with your insightful guidance. I have provided a table with each reviewer’s conditions and our response to these conditions, below. We have also submitted the reworked manuscript and a version of the manuscript with track changes highlighted, I hope this now meets with your approval.

All the very best

Dr Martin Clark 

Lecturer in Neurobiology 

University of Central Lancashire 

Preston, Lancashire 

PR1 2HE 

UK 

reviewer Reviewers comment Applicant response

Academic Editor 1. Please ensure your manuscript meets PLOS ONE’s style requirements, including those for file naming.

2. Please include captions for your supporting information files at the end of your manuscript and update any in-text citations to match accordingly. Thank you for your comments. We have now been through the document and amended it accordingly for both point 1 and point 2. You will find these formatting and style changes in the file named manuscript.

1 unclear as to why under 18's were omitted from the review, this would have enhanced the reach and value of the review especially as the increase in young people developing Diabetes type two has accelerated during recent years and has been associated with obesity

In Eligibility criteria (113) please explain there briefly your three word inclusion ( its in appendix) but this would have explained earlier to me why under 18's were not included and not all readers will be familiar with every guide Thank you for this comment. 

We have now added reference to why under 18’S were excluded earlier in the manuscript (Lines 134-136 of the track changes document) 

We completely agree that focusing on under 18’s would have certainly added value. We have added a comment in this regard in the discussion (Lines 456-459 of the track changes document). We greatly appreciate you highlighting this area where there is an apparent paucity of data. It clearly needs to be a focus of future primary research.

Thank you for highlighting this omission. We have no added explicit information on lines 126-128 of the eligibility criteria

1 

Was there any obvious correlation with Lockdown and any psycho social interventions associated with life style factors evident in any of the reviews 

Thank you for identifying this. This is clearly a very important point, which could have likely impacted the studies we reviewed. Thankfully the majority of studies we reviewed finished collating their data just prior to the pandemic. The publishing date of the reviewed articles seems to reflect more the impact the pandemic had on the writing and publishing processes. We have now added comment on this to the manuscript on line 419-424.

1 

Some of your methods were lengthy and whilst robust may have the effect of losing readers interest. 

Thank you once again for this comment. We have now been through the methods section (especially the study selection and data analysis sections) and edited to make more succinct. Please see the track changes document for evidence of this editing.

1 Small issues include where you note in text on page 3 (72)"In diabetes management research say then straight away such as eg Burns, 2020 etc. Likewise, (76) Accordingly a number of studies ie , (86) multiple studies ie

 Thank you, I have now rephrased the points Identified. Please see lines 73, 78 and 88. 

2 

Overall, it is recommended to re-read the manuscript and shorten it. 

It is recommended to summarize the results as table or graphical format, which will be attractive for both researchers and clinicians.

It is recommended to summarize the results in a table. Most readers are not interested in reading the text. 

Thank you for your comments. We have now been through the complete manuscript and edited for brevity as per your suggestion. This was extremely helpful, and we have managed to shorten the manuscript by approximately 2 pages. Please see the track changes document for evidence of this. 

Thank you. We greatly agree that graphical representation is extremely beneficial. Please see figures 2,3 and Table 2 for graphical representations of the results. 

Figures 2 and 3 were uploaded as. tiffs separately from the main manuscript as per PLOS one guidance. Table 2 can now be found embedded in the Manuscript file. 

We have also truncated the text related to these figures, as per your recommendations 

2 

Please end the introduction section with primary and secondary objectives instead of questions. It is not common in scientific writing. 

Once again, thank you for this comment. We have now amended as per your point. You can find this on line 98-106 of the track changes manuscript.

2 

Line 126- 130, you mentioned “Table 1 adapted from: Breuing J, Pieper D, Neuhaus AL, Heß S, Lütkemeier L, et 127 al. (2020) Barriers and facilitating factors in the prevention of diabetes type 2 and 128 gestational diabetes in vulnerable groups: A scoping review. PLOS ONE 15(5): 129 e0232250. https://doi.org/10.1371/journal.pone.0232250. Modified and licensed 130 under the creative common attribution 4.0 international (CC BY 4.0).” It better to cite to this reference.

Results:

- Line 195- 198 :” Fig 1. Adapted from: Page M J, McKenzie J E, Bossuyt P M, Boutron I, Hoffmann T 196 C, Mulrow C D et al. The PRISMA 2020 statement: an updated guideline for 197 reporting systematic reviews BMJ 2021; 372 :n71 doi:10.1136/bmj.n71. Modified and 198 licensed under the creative common attribution 4.0 international (CC BY 4.0)” Please cite to that reference

Thank you very much for this point which greatly improved the consistency of style. We have now added these references to the reference section and included the relevant number as part of the figure legend. 

The amendments can be found on lines 140 and 211 of the track changes document.

2 

How about searching for grey literature? 

Thank you very much for your comment. We have added a comment on line 160-161, which identifies that we searched for grey literature. We have also added a note on line 201-202 that clarifies that this search did not yield any additional results. We also felt it pertinent to add in the discussion that this grey literature search could have been more extensive, please see lines 455-456 for this comment. We do however feel that 

2 

In the abstract, you stated “ Studies included were of an interventional or observational design and restricted to English language”. However, in the method section, you did not mention anything about results synthesis of observational studies. 

Thank you for this most important point of clarity. I have now added a clarifying comment for this on lines 193-194 of the track changes document.

2 

Line 201-202, you included 36 studies. But, you stated that “Analysis of the 33 studies which were quantitative or mixed method revealed that 22 202 (67%) failed to include any psycho-social measures”. 

Thank you for this point of clarity. We have now added an explicit comment on lines 220-221 of the track changes document.

2 

Conclusion:

- Write this section based on the main research question.

Thank you for this. We have now adjusted the conclusion section accordingly.

---

## [Decision Letter · Decision Letter 1]

30 Oct 2023

Psycho-social factors associated with type two diabetes remission through lifestyle intervention: a scoping review

PONE-D-23-23765R1

Dear Dr. Clark,

We’re pleased to inform you that your manuscript has been judged scientifically suitable for publication and will be formally accepted for publication once it meets all outstanding technical requirements.

Kind regards,

Meisam Akhlaghdoust, M.D., M.P.H.

Academic Editor

PLOS ONE

Additional Editor Comments (optional):

Reviewers' comments:

Reviewer's Responses to Questions

**Comments to the Author**

1. If the authors have adequately addressed your comments raised in a previous round of review and you feel that this manuscript is now acceptable for publication, you may indicate that here to bypass the “Comments to the Author” section, enter your conflict of interest statement in the “Confidential to Editor” section, and submit your "Accept" recommendation.

Reviewer #1: All comments have been addressed

Reviewer #2: All comments have been addressed

2. Is the manuscript technically sound, and do the data support the conclusions?

Reviewer #1: Yes

Reviewer #2: Yes

3. Has the statistical analysis been performed appropriately and rigorously? 

Reviewer #1: Yes

Reviewer #2: Yes

4. Have the authors made all data underlying the findings in their manuscript fully available?

Reviewer #1: Yes

Reviewer #2: Yes

5. Is the manuscript presented in an intelligible fashion and written in standard English?

Reviewer #1: Yes

Reviewer #2: Yes

6. Review Comments to the Author

Reviewer #1: The questions that I posed have been answered, it would be interesting moving forward to do a follow on review that encompassed Covid and any online interventions that may have been or may subsequently be in place post Covid. I note that you acknowledge the gap in the under 18's studies and associated interventions, again as this is an area that is gathering momentum both from the increase in obesity and related diabetes type two diagnoses in younger people this area is something that would be worthy of following through as a second stage to the article that you have presented above.

Reviewer #2: I would like to thank the authors for addressing all the reviewer`s comments. There are no comments.

7. PLOS authors have the option to publish the peer review history of their article (what does this mean?). If published, this will include your full peer review and any attached files.

Reviewer #1: **Yes: **Dr.Toni Bewley

Reviewer #2: **Yes: **Mahin Nomali, Ph.D in Epidemiology, Tehran University of Medical Sciences

---

## [Editor Report · Acceptance letter]

7 Nov 2023

PONE-D-23-23765R1 

Psycho-social factors associated with type two diabetes remission through lifestyle intervention: a scoping review 

Dear Dr. Clark:

I'm pleased to inform you that your manuscript has been deemed suitable for publication in PLOS ONE. Congratulations! Your manuscript is now with our production department. 

Kind regards, 

on behalf of

Dr. Meisam Akhlaghdoust 

Academic Editor

PLOS ONE